# Integrated photonic metasystem for image classifications at telecommunication wavelength

Zi Wang [1], Lorry Chang[1], Feifan Wang[1], Tiantian Li[1] & Tingyi Gu [1✉]

Miniaturized image classifiers are potential for revolutionizing their applications in optical communication, autonomous vehicles, and healthcare. With subwavelength structure enabled directional diffraction and dispersion engineering, the light propagation through multi-layer metasurfaces achieves wavelength-selective image recognitions on a silicon photonic platform at telecommunication wavelength. The metasystems implement high-throughput vector-by-matrix multiplications, enabled by near $10^3$ nanoscale phase shifters as weight elements within 0.135 $mm^2$ footprints. The diffraction manifested computing capability incorporates the fabrication and measurement related phase fluctuations, and thus the pre-trained metasystem can handle uncertainties in inputs without post-tuning. Here we demonstrate three functional metasystems: a 15-pixel spatial pattern classifier that reaches near 90% accuracy with femtosecond inputs, a multi-channel wavelength demultiplexer, and a hyperspectral image classifier. The diffractive metasystem provides an alternative machine learning architecture for photonic integrated circuits, with densely integrated phase shifters, spatially multiplexed throughput, and data processing capabilities.

[1] Department of Electrical and Computer Engineering, University of Delaware, Newark, DE 19711, USA. ✉email: tingyigu@udel.edu

Enabled by the subwavelength structures, metasurfaces are capable of high spatial resolution phase control, photon momentum steering, and high-efficiency diffraction[1–3]. Designed dispersion and diffraction enable multi-layer metasystems for powerful optical analog signal processing[4–15]. The metasystems can perform mathematical operations of the impinging electromagnetic wave with subwavelength resolution[16–20]. Fourier transform method designed passive metasurface systems demonstrate real-time spatial differentiation and edge detection[21–29]. Beyond the deterministic functions, spatial information classifications are also demonstrated in free-space optical systems. Designed by deep neural networks, multi-layer free-space diffractive optic elements perform high-accuracy image classification and logic computing in millimeter and micrometer wavelength ranges[30–34]. With only one layer of phase or amplitude coded mask, the light flow in the free-space 4F system mimics the matrix calculation in convolutional neural networks, achieving high accuracy image classifications with computer-aid postprocessing[35,36]. A diffractive processing unit based on paired spatial light modulator-CMOS sensor arrays implements powerful deep learning tasks, with intermediate optoelectronic signal conversions for signal flow between the optical units[37]. In those optical systems, the pixel or cell size in the coded plane is well beyond wavelength, which sets fundamental limitations on the diffraction efficiency and spectral engineering capability.

Metasurface-based multi-layer systems, named metasystem, expand the functionality of metasurface in the out-of-plane dimension[38–40]. Lithographically assisted alignment and bonding between metasurface layers are required for providing sufficient precision and robustness in functional metasystems[39,40]. The integrated photonics platform provides such alignment with one-step lithographically defined multiple metasurface layers. Compared to the waveguide-based integrated photonic processors[41–43], the metasystem architecture offers higher throughput vector-by-matrix multiplication, which can be further expanded by wavelength-division multiplexing (Supplementary Note 1)[44,45]. The metamaterial manifested weight element density, combined with diffraction strengthened inter-layer connectivity, enables the passive system to accomplish machine learning tasks of spatial pattern classification (Fig. 1a). The diffraction manifested data processing capacity allows the training process to incorporate the random phase offsets caused by nanofabrication and measurement. Unlike the other integrated photonic processors[41–43], the passive photonic metasystems are fully functional without active layers for phase correction. The passive integrated metasystem can grasp the key information with a femtosecond single-shot exposure, and thus save the time and energy consumption for subsequent electronic processing for on-the-fly data compression.

As the depth of a machine learning system outweighs the number of elements per layer, here we demonstrate high accuracy image classifications at telecommunication wavelengths in the multi-layer one-dimensional metasurface systems. Arrays of rectangular slots are defined in the silicon layer. The slot lengths in those phase-only transmissive arrays are pre-trained by deep diffractive neuron networks. Beyond conventional classification functions, the metasystems also demonstrate unique functions of wavelength demultiplexing and multi-wavelength pattern classifications, with potential applications in spatial division multiplexing based optical interconnects and machine vision[34,46].

## Results

**Design of the metasystem.** The metasystems are defined on a silicon on insulator (SOI) substrate with single-step lithography

and dry etch (Method). As individual cells in metasurface, the geometry of the rectangular slots are learnable parameters. Each pair of the slots represents a weight element and connects to the following layers through diffraction and interference of the in-plane waves (Fig. 1a, b). Both amplitude modulation and the phase shift of the transmitted wave can be programmed by adjusting the width and length of the subwavelength slots, respectively[47,48] (Supplementary Fig. S2). With a fixed slot width of 100 nm and lattice constant of 500 nm, the phase shift of the transmission coefficient can be continuously tuned from 0 to $2\pi$ with the slot length, while the amplitude stays more than 95% (Fig. 1c). Figure 1d shows the angle-dependent complex transmission coefficient. The amplitude of the transmission reduces to half as the incident angle increases from 0° to 28°, with phase distortion less than 0.1 (in the unit of $2\pi$ rad). The results in Fig. 1d are insensitive to the slot length (Supplementary Fig. S3). Distinguished from our prior demonstration of gradient metasurface-based mathematical operators, large phase contrasts between neighboring cells are required in the metasurfaces for machine learning tasks. As the transmission coefficient of each metasurface design is numerically calculated from a periodic array, single-slot implementation of each phase shifter in gradient metasurface design results in unexpected discrepancies, and thus two subwavelength slots are employed here for representing one phase shifter in the designed network[48] (Inset of Fig. 1d).

The diffractive metasystem is firstly designed in Python and then verified by finite-difference time-domain (FDTD) simulations and experiments. During the training stage, the phase shifts in each metasurface layer are iteratively updated by following the gradient descent algorithm (Supplementary Note 2)[49]. In the forward propagation step, we calculate outputs of the metasystem with input data from the training dataset. The difference to target outputs (the loss) is then derived for the next step. In the backpropagation step, we calculate the gradient of the phase for every cell and then update the phase value to decrease the loss. The random phase offset with uniform distribution within the interval $[0,0.5\pi]$ are introduced to each cell during the training stage, to improve the system's robustness against nanofabrication variations and free-space phase fluctuations in measurement (Supplementary Fig. S1a). The photon propagation from layer $l$ with $k$ neurons to the next layer with $n$ neurons resemble the vector-matrix multiplication:

$$\left[ m^{l+1}(1), \quad \ldots, \quad m^{l+1}(n) \right] = \left[ m^l(1) * t^l(1), \quad \ldots, \quad m^l(k) * t^l(k) \right] \cdot W \tag{1}$$

where $t^l(p) = a * \exp[j\phi^l(p)]$ represents the transmission coefficient of the $p$-th neuron in $l$-th layer. The amplitude $a$ is near 1 for the slot width of 100 nm. The phase shift $\phi^l(p)$ is proportional to the slot length. $m^l(p)$ and $m^{l+1}(q)$ are the amplitude of input photons towards the $p$-th neuron in the $l$-th layer and the $q$-th neuron in the $(l+1)$-th layer, respectively. The inter-layer connectivity $W$ is a $k \times n$ transfer matrix derived by the Rayleigh–Sommerfeld diffraction equation, representing the wave propagation in the SOI slab waveguide (Fig. 1b). The $(p, q)$-th element of the $W$ is:[50]

$$w(p, q) = \frac{\triangle y}{r^2} \left( \frac{1}{2\pi r} + \frac{1}{j\lambda} \right) \exp\left( \frac{j2\pi r}{\lambda} \right) \tag{2}$$

where $r$ is the distance between the $p$-th neuron in layer $l$ and the $q$-th neuron in layer $l+1$. $\lambda$ is the effective wavelength in the planar waveguide. Considering the angle-dependent transmission amplitude (Fig. 1d), an additional factor of $U(\Delta y) \propto e^{-\left[\frac{\pi \Delta y \sigma}{\lambda a}\right]^2}$ is superimposed onto the outputs of each layer, where $\Delta y$ is the relative distance along the $y$-direction, $a$ is the spacing along the $x$-direction. $\sigma$ is 0.45 μm for the first layer and 0.08 μm for the

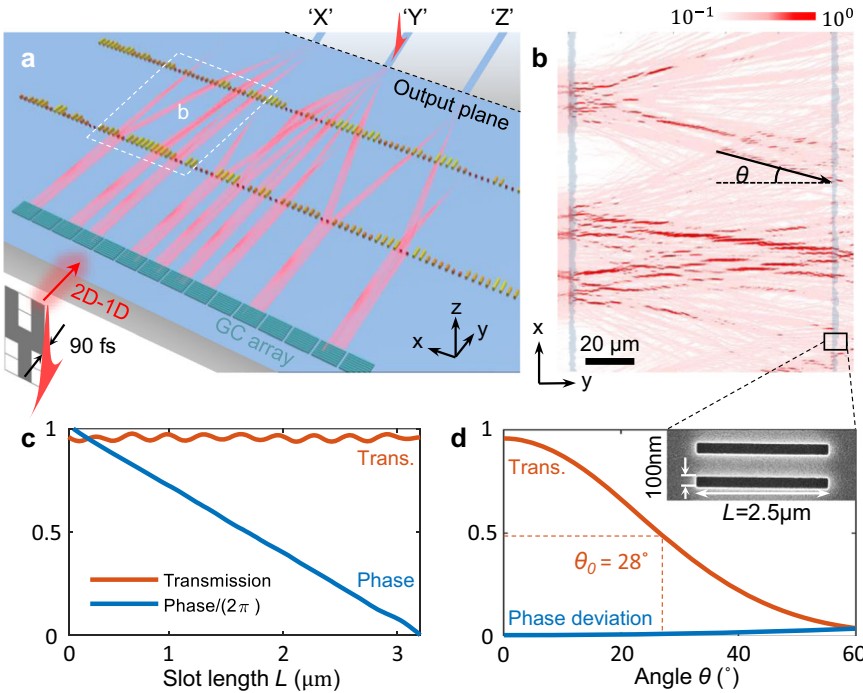

**Fig. 1 Integrated metasurface system for spatial pattern classification. a** Schematics of the system architecture. As an example, a pattern 'Y' with less than 90 femtosecond pulse duration is coupled to the integrated metasystem through the input grating coupler array (green gratings). Through the diffractions through the metasystem, the light is converged onto the position on the output plan where the correspondent waveguide channel locates. **b** Subwavelength structure manifested diffraction between metasurface layers. The numerically simulated light intensity is superimposed onto the optical microscope image of a fabricated metasystem. **c** Amplitude and phase of the complex transmission coefficient versus the slot length $L$ (indicated in the inset of **d**) with the fixed slot width of 100 nm. **d** Amplitude and phase of the complex transmission coefficient versus the incident angle ($\theta$ indicated in **b**) with the slot width of 100 nm and $L$ of 2.5 μm. Inset: Scanning electron microscope (SEM) image for the zoom-in view of the metasurface cell in **b**.

subsequent layers, obtained by fitting the model to the numerical simulation results.

**Metasystem for spatial pattern classification.** As an example, we implement an integrated two-layer metasystem for letter classifications. Each metasurface layer contains 450 phase shifters. The inter-layer distances are selected to be 100 μm, balancing the insertion loss and classification accuracy (Discussion section). The metasystems and grating couplers are defined on the SOI substrate with single-step lithography and etching process (Methods). The setup for characterizing the metasystem is illustrated in Fig. 2a. The input patterns are reshaped from a two-dimensional (2D) matrix to a one-dimensional (1D) vector and then projected onto the 1D grating coupler array through a digital micromirror device (DMD). The input patterns are the binary letter images with 15 pixels (bottom insets in Fig. 2a). The outputs are collected by a single-mode fiber through a grating coupler and delivered to a broadband infrared (IR) photodiode. A digital IR camera monitors the alignment between the reflected patterns from DMD and the grating coupler array. The optical image (left in Fig. 2a) shows the perspective view of one device under test (DUT). A single mode fiber picks up the signal from output grating couplers on DUT (Fig. 2b). The scanning electron microscope (SEM) images show the detailed nanostructures of the grating coupler array (Fig. 2c) and the pre-trained metasurface (Fig. 2d) on DUT.

The testing dataset is the binary letter images with amplitude flipping in random pixels (Supplementary Fig. S1b). The two-layer metasystem is pre-trained by 10,000 such matrices. Numerical testing by the other 1000 datasets predicts 98%

accuracy in letter classifications. Figure 3a shows an example optical field intensity distribution of the optical diffractive network. Three waveguides are placed 100 μm apart on the output plane, representing three channels of classification results. Channels 1,2 and 3 are correspondent to the input letter patterns of 'X', 'Y', and 'Z' respectively. With an input image of the letter 'X', the light intensity is significantly higher near the spatial position of channel 1 (Ch1) on the output plane. The detailed light intensity distribution along the output plane is plotted as gray lines in Fig. 3b. The experimentally measured data (squares with error bars) are consistent with FDTD simulations (Fig. 3b). The blue, red, and yellow squares are the light intensity from the grating couplers connected to Ch1, 2, and 3, respectively. At 1550 nm continuous wave (CW) input, numerically simulated (Fig. 3c) and measured confusion matrix (Fig. 3d) show the classification accuracy of 96% and 92%, respectively. The metasystem's response is consistent for the CW inputs across the C and L bands (Supplementary Note 3). The broadband operation is critical for ensuring high classification accuracy of single-shot ultrafast pulsed inputs. Under 90 femtosecond pulsed light (centered at 1551.6 nm with a bandwidth of 30 nm), the measured confusion matrix shows 89% classification accuracy in this metasystem (Fig. 3e). Numerical simulation shows the insertion loss in the metasystem classifier is 9.3 dB.

**The dispersion engineering of the metasystem.** The dispersion of the metasurface system can be tailored for expanding device applications to machine vision and hyperspectral imaging. To show the spectral engineering capability, we implement a three-layer metasystem that can effectively separate input signals

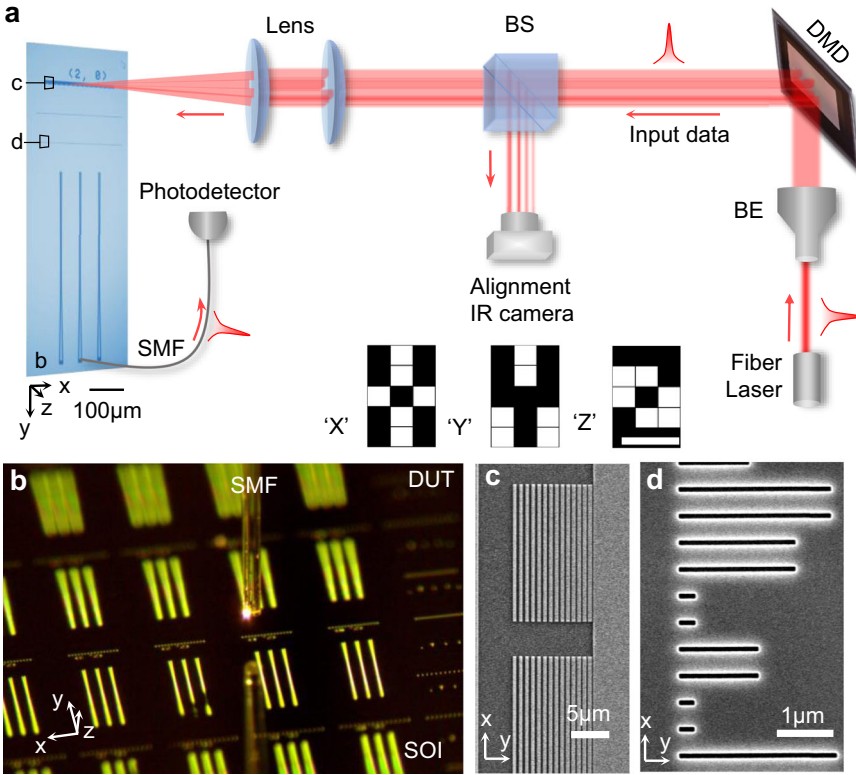

**Fig. 2 Device characterization. a** The schematic illustration of the confocal set-up. The infrared laser beam is firstly expanded by the beam expander (BE). The reflection from the programmed micromirror device (DMD) carries the input pattern. The reflected image is then focused on the input grating couplers arrays. The output signal is coupled to a single-mode fiber (SMF) through a grating coupler. An IR camera is used to monitor the light coupling on and off the chip through a beam splitter (BS). The bottom insets are the input images for the letter 'X', 'Y', and 'Z'. **b** The device under test (DUT) includes arrays of metasystems fabricated on a silicon-on-insulator (SOI) substrate. **c** SEM images show the nanostructure of the input grating coupler array and **(d)** metasurface structures.

centered at 1490, 1530, and 1570 nm (Fig. 4). The distances between the input plane, metasurfaces and output plane are fixed at 100 μm. Three parallel output waveguides are spaced 30 μm apart along the $x$-direction. Under CW tunable laser excitation at 1490 nm, light merges at the destinated $x$-position on the output plane, where channel 1 waveguide locates (Fig. 1a). The numerically predicted spectra along the output plane (Fig. 1b and dashed curves in Fig. 4c) align with experiments collected from the three output channels (solid curves in Fig. 4c). The blue, red, and orange curves represent the outputs for channels 1, 2, and 3 respectively. The measured insertion losses for such a three-layer system are 13.1 dB, 16.8 dB, and 18.9 dB for the wavelength at 1490 nm, 1530 nm, and 1570 nm, respectively. The spectral resolution of such a metasystem is limited by the number of output ports. The spectral resolution of 7 nm can be achieved with 11 output ports.

The complicated diffraction and interference allow one-to-one correspondence between the spatial distributions of the light and the laser wavelength[51]. Combining both features, we design and experimentally demonstrate a two-wavelength pattern classification system (Fig. 5). An optical image of the designed metasystem is shown in Fig. 5a. The input grating couplers design is same as the one in Fig. 3. The metasystem is composed of 2-layer metasurfaces with 600 phase shifters per layer. The 6 output ports are correspondent to pattern "X", "Y", and "Z" at 2 input wavelengths of 1530 nm and 1570 nm. For the input pattern of "Y" at 1570 nm, the simulated light distributions on the output plane (gray curves in Fig. 5b) are consistent with measured data points (solid squares in Fig. 5b). The measured confusion matrix

(Fig. 5c) indicates the hyperspectral pattern classification accuracy of 70%, with an insertion loss of 14.2 dB.

## Discussion

Compared to the 2D metasystem in free space, the metasurface on the integrated platform is limited to a smaller number of cells and out of plane-in plane couplers, with the advantages of lower insertion loss and feasible fabrications for multi-layer structures. With the same total cell number, classification accuracy is more sensitive to the depth of metasystems than the size of each layer (Supplementary Fig. S7a). Currently, the fabrication limited metasurface cell number is $10^4$, which is sufficient for the standard testing databases with propagation matrix compression (Supplementary Note 2.2). We numerically explored the 1D metasystem's computing capability by designing one for a Modified National Institute of Standards and Technology (MNIST) handwritten digit database with 784-pixel inputs (Supplementary Note 4). 3 Epochs bring a metasystem's accuracy to be 96% (Supplementary Fig. S6). Currently, the main technical challenge is the layout design of a large number of I/O ports on an integrated photonic platform with tolerable phase distortions from nanofabrication. Theoretically, a 2D metasurface with the sub-wavelength cell owns significant computing capabilities. However, experimental implementation of such a system for machine learning has never been reported in telecommunication wavelength or infrared, but feasible if the fabrication or alignment errors are considered in the training process (Supplementary Note 2.1, 2.5). Commercially available components (DMD or diffractive optical elements) have a typical cell number of $10^4$-$10^6$.

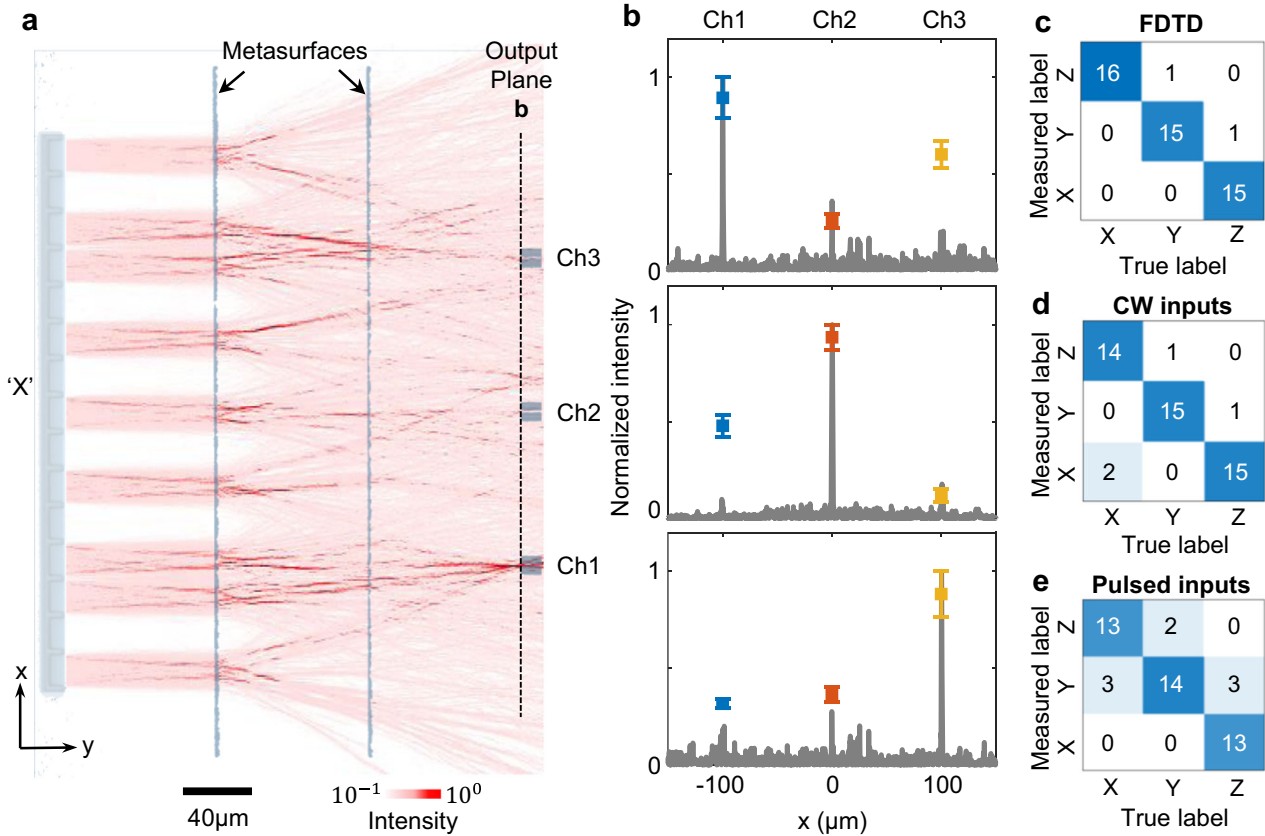

**Fig. 3 Broadband pattern classifier. a** The in-plane electric field intensity distribution with the input patterns of 'X' (log scale), superimposed onto an optical microscope image of the correspondent metasystem device. **b** Comparison of measured optical intensities on the three waveguides placed on the output plane (dots with error bars) and the numerically simulated optical distribution on the output plane (gray curve). The error bars represent the standard deviation (s.d.) for 16 measurements. **c** Confusion matrix for numerical simulation and **d** measured results under CW excitation, with a center wavelength of 1550 nm and bandwidth of 2 pm. **e** Measured confusion matrix for a 90-femtosecond light source (centered at 1551.6 nm with a spectral bandwidth of 30 nm).

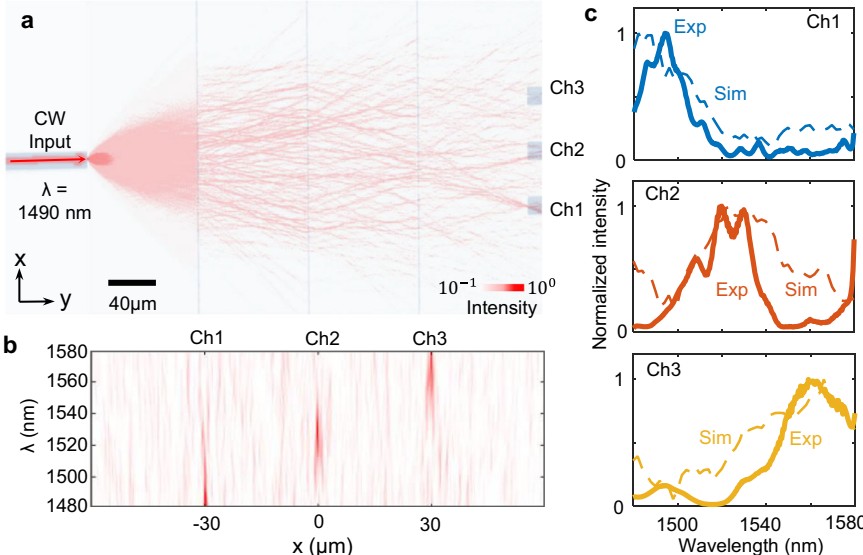

**Fig. 4 Dispersion engineered three-layer metasystem for wavelength identification. a** The simulated in-plane light distribution is superimposed on the optical image of a fabricated device. With an input wavelength of 1490 nm, light paths merge near the position of channel 1 on the output plane.
**b** Measured (solid lines) and simulated (dashed lines) spectra at three output ports. **c** Simulated optical intensity distributions on the output plane. Three waveguide output ports (Ch1, Ch2 and Ch3) are centered at x = −30, 0 and 30 μm respectively.

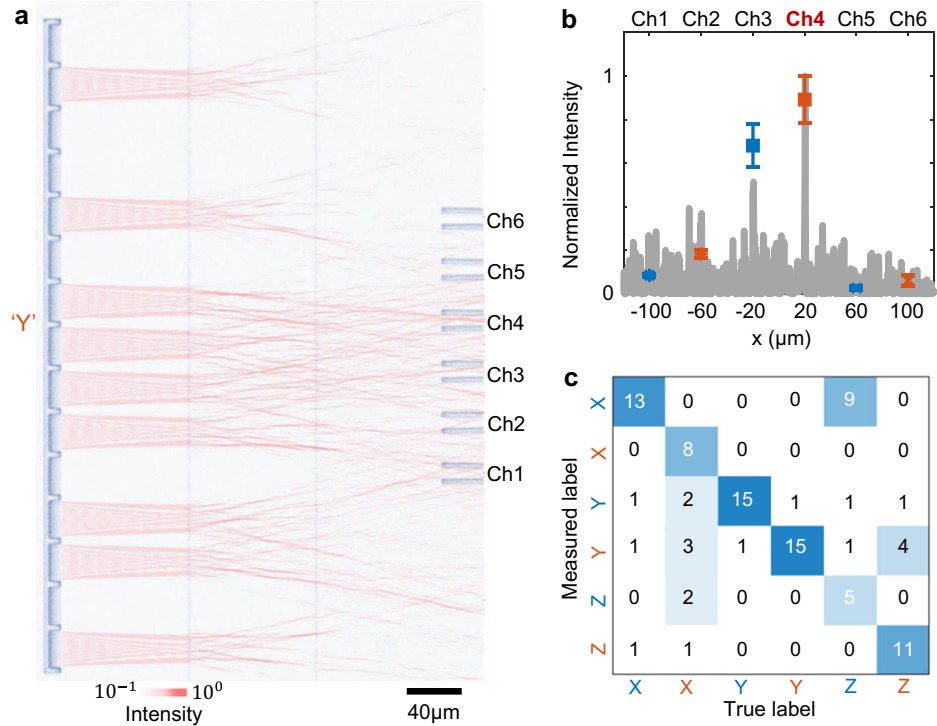

**Fig. 5 On-the-fly hyperspectral image classifier. a** The simulated in-plane light distribution with the input pattern "Y" at 1570 nm is superimposed on the optical image of a fabricated device. **b** Measured signals on 6 channels for the same input as a (squares with error bars), compared to the simulated distribution at the output plane. The error bars represent the s.d. for 16 measurements. **c** The confusion matrix of the experimental results. Blue and red letters indicate the incident wavelength of 1530 nm and 1570 nm, respectively.

Single layer component has been utilized for high-accuracy image classifications[35,36]. The integrated photonic platform can eliminate out-of-plane light diffraction, and thus result in orders of magnitude lower insertion loss compared to free-space optical systems.

Based on the Toeplitz matrix, the training algorithm of the 1D metasystem requires less memory and time during the training process (Supplementary Note 2). The time and computational cost-efficient design algorithm facilities systematic design studies of the MNIST classifiers (Supplementary Note 4). Given sufficient weight matrix size, a one-layer metasystem can only achieve 88% accuracy. 5–10% accuracy boost is observed with increased layer number (Supplementary Fig. S7a). The diffraction distance is proportional to inter-layer distance, which results higher classification accuracy and insertion loss (Supplementary Fig. S7b).

The reconfigurability and nonlinear activation functions can be introduced into the metasystem platform via hybrid integration of active materials. For example, phase change materials with high refractive index contrast can fill those slots and provide sufficient phase tunability[52] for a fully reprogrammable metasystem. Certain active materials exhibit exceptionally high nonlinear responses (such as two-photon absorption-related free carrier absorption or absorption saturation) and are transparent at telecommunication wavelength ranges, which can be integrated into the diffractive networks as nano-scale activation functions with solution processing[53,54].

Designed by diffractive optical networks, we experimentally demonstrate cascaded metasurface systems for wavelength-selective pattern classifications in telecommunication wavelength. The miniaturized metasystem is fabricated on SOI substrate with one-step lithography and etching. Compared to conventional integrated photonic circuits, the manifested throughput and computing capability in the metasystem is attributed to dense phase shifters and efficient diffractions. With proper training, the integrated metasystem can be robust against input noise and random nanofabrication offsets. As a spatial pattern classifier, 92% and 89% accuracy are achieved in a two-layer metasystem, under narrow-band CW excitation and broadband femtosecond pulse excitation, respectively. The broadband operation of the pattern classifier allows single-shot image classification with boosted parallelism for optical signal processing. The wavelength selectivity of such a metasystem can be co-designed with the pattern classification function for hyperspectral imaging, machine vision, and hardware accelerators.

## Methods

**Device fabrication**. The integrated metasystem is fabricated on an SOI substrate from Soitec, with a 250 nm silicon layer and a 3 μm thermal dioxide layer. The designed patterns of the metasurface, waveguides, and grating couplers are firstly defined in CSAR 6200.09 positive resist layer by using a Vistec EBPG5200 electron beam lithography system with 100 kV acceleration voltage, followed by optimized resist development and single-step dry etch procedures. A 300-nm thick silicon dioxide protection layer is finally deposited on the sample by plasma-enhanced chemical vapor deposition (PECVD). The loss of grating couplers and channel waveguides used in the devices are less than 6 dB and 1 dB respectively.

**Optical measurements**. Tunable lasers (ANDO AQ4321A and AQ4321D) generate coherent and linearly polarized light with 1 pm spectral resolution. For the pulsed signal measurement, a femtosecond laser centered at 1551.6 nm with a duration less than 90 fs and spectral bandwidth around 30 nm (Calmar laser CFL-10CFF) is used to replace the continuous wave light source. The infrared light travels through a polarization controller, a beam expander, DMD (Texas Instruments DLP650LNIR), a lens, a long working distance objective (a Mitutoyo Plan Apo 20× infinity-corrected objective), and incident onto the input grating couplers. A single-mode fiber probe collects optical outputs and sends them to an InGaAs photodiode and optical power meter (Newport 818-IG-L-FC/DB and 1830-R-GPIB). A 640 × 512-pixel format and 25 μm pitch size digital IR camera (Goodrich SU640KTSX) monitors the input pattern alignment with the substrate.

**Numerical simulations.** The integrated optical diffractive network is constructed in the PyTorch framework (Supplementary Note 2)[55] and verified by the 2D FDTD method.

## Data availability

The datasets generated during the current study are available in the Zenodo repository, https://doi.org/10.5281/zenodo.6345622.

## Code availability

The python script used in this paper have been deposited in the Zenodo repository, https://doi.org/10.5281/zenodo.6339743.

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

## Acknowledgements

This work was supported by AFOSR Young Investigator Program (FA9550-18-1-0300) and an Early Career Faculty grant from NASA's Space Technology Research Grants Program (80NSSC17K0526). The devices are fabricated at the University of Delaware Nanofabrication Facility with assistance from Dr. Kevin Lister.

## Author contributions

Z.W. and T.G. conceived the idea. Z.W. developed the design principle and performed numerical simulations. Z.W. and T.L. fabricated the samples. Z.W., L.C, and F.W. built the experimental set-up. Z.W. performed the measurements and analyzed the data. Z.W. and T.G. wrote the manuscript with inputs from all authors.

## Competing interests

The authors declare no competing interests.
