## [Peer Review File · Nature Communications]

Integrated photonic metasystem for image classifications at telecommunication wavelengthREVIEWER COMMENTS

Reviewer #1 (Remarks to the Author):

The authors of this manuscript presented an image recognition by a passive silicon photonic metasystem, and nearly 90% accuracy with femtosecond inputs was achieved. The authors demonstrated a one-dimensional network based on their previous work (Ref. 36). The manuscript contains some new results, but I have important concerns on the novelty and significance of the manuscript.

1. Passive optical structures for neural network computing have been verified by many groups including one- or two-dimensional metasystems with all-dielectric [Ref. 25][Ref. R1]. In the authors' previous work, they have achieved wave-front manipulations with the similar structure. Thus the matrix computing with the phase manipulation can be certainly realized. Although the authors have obtained good classifying performance, the passive structure can hardly handle different training targets. The authors should investigate a more powerful and convincing innovation point.

[R1]: Chang, J., Sitzmann, V., Dun, X., Heidrich, W. & Wetzstein, G. Hybrid optical–electronic convolutional neural networks with optimized diffractive optics for image classification. *Sci. Rep.* 8, 12324 (2018).

2. The reported benchmark tests are very simple. The forward structure for the proposed deep diffractive neuron networks is the same as that in Ref. [25]. As the title claims, the purpose of the proposed integrated photonic metasystem is to conduct image classification tasks. However, compared to Ref. [25], the image classification cases demonstrated in this manuscript are much simpler. I suggest to demonstrate a more interesting image classification case to enhance the novelty of the manuscript.

3. Recently, nonlinear activation function is common in the optical neural networks. But the proposed integrated photonic metasystem is still a linear one.

Some minor issues:

4. In Fig. 2b, I notice that some values of the normalized intensity is above 1 according to the error bar.

5. Some grammatical mistakes and typos are found, and hence the language quality needs to be improved.

Reviewer #2 (Remarks to the Author):

By using the deep diffractive neuron networks trained silicon metasystem, the authors demonstrate the function of image recognition. The metasystem can do high-throughput vector-by-matrix multiplications, enabled by the passive subwavelength phase shifters. A 15-pixel spatial pattern classifier can reach near 90% accuracy. The metasystem can also potentially improve the data processing capability of the CMOS compatible photonic integrated circuits. The miniaturized image classifiers have great applications potentials in optical communication, autonomous vehicles, and healthcare. This is a very interesting work; the experimental results agree with the numerical simulations. Therefore, I would like to support its publication after revisions.

Comments:

1. The optical efficiency of the metasystem should be discussed in more details.
2. Figure 1a, the phase shifters have low image resolutions, I would suggest using less shifters to show the concept. Figure 1c and 1d, it is not easy for the audience to understand. The basic configuration of incident plane of light should be depicted.
3. Figure 2d, for the non-expert, it is difficult to understand the working principle. For example, the SMF is used to collect the light, but people can not see where is the excitation. It is better to re-plot Figure 2a and 2d. The structures on the DUT are not clear as well.
4. In the discussion part, the authors mention that “compared to the 2D metasurface, the number of the elements is significantly reduced”. This is true, however, I would expect the efficiency and the quality image recognition by using a 2D metasurface is even better. Please explain more about this.
5. Many important literatures in the field of metasurfaces are not included in the references.

Reviewer #3 (Remarks to the Author):

The manuscript under review deals with the realization of an optical analog signal processor to enable image classification in the infrared range. The diffractive nature of the proposed platform comprising of

cascaded 1D metasurfaces somehow mimics the functionality of a trainable artificial neural network. Such a passive metasystem provides acceptable performance measures including classification accuracy, throughput, and footprint. Overall, this work is a timely contribution to the on-demand topic of biologically inspired optical computing. The manuscript is organized, and the results look sound. I believe that this manuscript can stimulate further research lines in visual computing, image analysis, and feature detection. Said that, I think that the work is important enough to be published in Nat. Commun. However, there are several major concerns/suggestions that need to be addressed before I can recommend its publication.

- There exist two major concerns with optical neural networks (ONN): i) lack of nonlinearity (mimicking the nonlinear activation function of a neuron) which limits the functionality of ONN to a simple linear transformation of the incident light, ii) absence of tunability which hinders the online training of neurons. This hinders the realization of biologically inspired computing paradigms. Can the authors comment on how their platform can address these issues?

- The design principle of the metasurfaces is based on the arrangement of length-variant slots (Fig. 1c). However, Fig. 3a clearly shows that all slots have the same lengths. Can the authors comment on this?

- Given that in one direction the length of building blocks is up to 3 μm , can we still call “subwavelength phase shifters”? This brings a more fundamental question to the stage; can we call the proposed structure a “metasystem”?

- I assume that the authors adopted the same configuration in [Nature Commun. 10.1 (2019): 1-7], in which the periodicity is considered 500 nm. Considering Fig. 3a, the length of the metasurface (in the x-direction) is $\sim 130 \mu\text{m}$ which means that ~ 260 meta-atoms are effectively in charge of signal processing. This is way lower than the 450 reported in the main text and Table S1. Can the authors comment on this discrepancy? If this is the case, then almost half of the slots are idle and have no contribution to the calculated 2×450 weight matrix.

- The authors claimed that their classifier is robust against the fabrication imperfections. While this is inherently afforded by large-scale metasurfaces, it needs, at least in terms of FDTD simulations, to be justified. Confusion matrices for different scenarios can be provided.

- Given 90-fs-long pulses, how the authors reached a throughput of 1015 b/s? Above that, considering the long-distance dispersive integrated platform, what is the delay (and corresponding speed) associated with the optical processing? Did the author consider this factor when calculating 1000 Tb/s throughput? Can the authors comment on the speed of the photodetector?

- The 1D nature of the proposed platform (compared to free-space metasurfaces) highly limits the classification of high-resolution images. The authors demonstrated 15-bit inputs. Can the authors discuss this limitation and compare it with 2D metasystems?

- I could not find Ref. [3] related to the optical signal processing as it mainly discusses the nanophotonics design using deep learning approaches. Also, the realization of mathematical operations using metamaterials is not covered in Refs. [14, 15] as mentioned in the introduction section. I found the following works more related: [Nat. Commun. 8, 15391 (2017)], [Nano Lett. 15.1 (2015): 791-797], and [Optics Lett. 40.22 (2015): 5239-5242].

- The authors motivated their work by referring to the possible challenge of interlayer misalignment of multilayered metasurfaces. However, several works (e.g., [Optica 7.1 (2020): 77-84] and [Nano letters 18.12 (2018): 7529-7537]) experimentally demonstrated that such an issue can be mitigated through highly accurate bonding processes or integration of vertical heterostructures. Can the authors provide more motives/rationales for the proposed architecture?
- Can the authors comment on why the phase distortion with oblique incidence is negligible?
- “Both amplitude modulation the phase shift of the transmitted wave can be programmed by adjusting the width and length of the subwavelength slots (Fig. 1c-d) [36].”, While Fig. 1c shows the variation of amplitude/phase as a function of the length of the slot, Fig. 1d represents this measure versus the angle of incidence. It would be better to provide such measures as a function of both width/length in the Supplementary and revise the text accordingly.
- The width of the meta-atom in Fig. 1C and width/length of the meta-atom in Fig. 1D should be mentioned in the caption. Can the authors get similar curves for different slot lengths by changing the angle of incidence?
- While the first layer of the metasurface accepts a small angle of incidence, this is not the case for layer 2 (and 3). It means that the amplitude variation across the intermediate layers is not negligible. This highly affects the vector-matrix multiplication formula governing the operation of DNN. Can the authors verify this?
- “The input grating couplers are one-dimensional (1D) gradient metasurface and about 10 μm wide in x direction (Fig. 2d).”, to my eye, the width of grating is $> 13 \mu\text{m}$. Can the authors report the exact values? Also, adding an optical image of a single device with enlarged SEM images of building blocks can be more illustrative.
- Figure 3b has different scales for simulation and measured data. If the measured optical intensities are normalized why the error bar exceeds 1? Can the authors comment on the normalization strategy?
- “Each phase shifter in the hidden layer is represented by two subwavelength slots ...”. Based on Fig. 1, the design methodology is based on formation of a single slot. Where does this discrepancy come from?
- To improve the system’s robustness against nanofabrication, random phase noise in the interval of $[0, 0.5\pi)$ is added during the training process. What is the reason behind selecting this interval?
- The bandwidth of the femtosecond laser in the main text is given “near 20 nm” while in the caption of Fig. 3 is “over 30 nm” and in the Methods is “around 50 nm”. Consistency is important throughout the manuscript.
- Why is the interlayer distance chosen 100 μm ? Why is 30 μm spacing considered in the output?
- According to the confusion matrix in Figs. 3d, the classification accuracy for CW is $\sim 94.4\%$.
- In my view, Table S1 is not accurate. The signal format for Refs. [S2, S3] are temporal-spectral, the signals are coherent, and their architectures are based on MZIs and directional couplers, respectively. I recommend the authors revise the table according to the reported data provided in those papers. Also, the footprint is given 1 mm^2 in the abstract while reported 0.045 mm^2 in the table. Moreover, the

number of phase shifters in the abstract is 103 while in the table is 2×450 . Finally, WDM stands for wavelength division multiplexing that needs to be corrected!

- The authors did not discuss the optical performance of their architecture. This is an important performance measure (specifically for scalability) which needs to be compared with other PIC architectures.

- Why are the output ports in Fig. S2 100 μm away from each other?

- The device architecture presented in the main text has two layers of metasurfaces. For more consistency, I recommend revising Fig. 1a accordingly.

- According to the scale bar in Fig. 2e, it seems the length of some slots is $> 3.7 \mu\text{m}$? Is there any specific reason behind this, and how this affects the overall response? It would be good to consider this when generating Fig. 1c. Also, the width of slots seems to me $\sim 130 \text{ nm}$. This is not in accordance with the simulation results.

- Most figures can be regenerated with higher resolution and details.

- Among minor concerns, there are some (though not many) typos and poorly formulated sentences across the manuscript:

* Page 2: "Both amplitude modulation the phase shift", "Fig. 1a-b", "meta-system", "represents a weight element and connecting"

* Page 3: "Fig. 1c-d", "gradience descent algorithm"

* page 4: (Fig. 3a) is duplicated in one sentence, Fig. 2c should be revised as Fig. 2b, "With an input image of letter 'X', the light intensity onto channel 1 (Ch1) on the output plane", "The light intensity the position of Ch1", "10,000 such datasets", "1000 testing dataset", "and experimentally verified scanning the input"

* page 5: "by a designing one for"

* Page 6: "complexed", "in such system", "device layer", "e.g."

* Page 10: "coupled onto"

Responses to Reviewers' Comments

Dear Editor,

We are responding to your letter regarding our manuscript (**ID: NCOMMS-21-16949, Title: Integrated photonic metasystem for image classifications at telecommunication wavelength**). Thanks for the reviewers' patience. We have carefully considered the reviewers' comments and revised our manuscript accordingly. Meanwhile, we designed, fabricated new devices and added new experimental results to strengthen the paper. All the revisions/additions are highlighted in the manuscript and identified in the correspondent response in **red**.

Reviewers' comments:

Reviewer 1:

The authors of this manuscript presented an image recognition by a passive silicon photonic metasystem, and nearly 90% accuracy with femtosecond inputs was achieved. The authors demonstrated a one-dimensional network based on their previous work (Ref. 36). The manuscript contains some new results, but I have important concerns on the novelty and significance of the manuscript.

1. Passive optical structures for neural network computing have been verified by many groups including one- or two-dimensional metasystems with all-dielectric [Ref. 25] [Ref. R1]. In the authors' previous work, they have achieved wave-front manipulations with the similar structure. Thus the matrix computing with the phase manipulation can be certainly realized. Although the authors have obtained good classifying performance, the passive structure can hardly handle different training targets. The authors should investigate a more powerful and convincing innovation point.

[R1]: Chang, J., Sitzmann, V., Dun, X., Heidrich, W. & Wetzstein, G. Hybrid optical–electronic convolutional neural networks with optimized diffractive optics for image classification. *Sci. Rep.* 8, 12324 (2018).

Response: Thanks for the reviewer's comments and the reference paper. In our paper, we only discussed the comparison to the multi-layer deep neuron network designed system but did not include the 4f Fourier optic system-based optical convolutional networks (CNN). In the reported

CNN system, only one layer of the mask was used, which avoids the pixel-to-pixel alignment as deep neuron network-based system needs. We have included related works in the introduction part and added a comparison table for related image classifiers as tables S2.

The fundamental difference between the presented metasystem architecture and the CNN-based bulk system is the subwavelength ‘pixel’ (defined as slot in our device). The subwavelength design allows large-angle light diffraction and dispersion engineering. A large diffraction angle empowers computing capability and connectivity between the layers. As the diffraction angle is inversely related to the aperture size, a few-pixel gradient metasurface allows full-vector steering of the normal incident wave, as we demonstrated in our previous work [original reference 25, updated reference 48]. The dispersion engineering is unique in such a metasystem, but not feasible in bulk optical systems.

This work is the first demonstration of a passive metasurface system (metasystem) for machine learning tasks at telecommunication wavelengths (on both free-space and on-chip platforms). As each component in the metasystem relies on nanofabrication, its reliability, reconfigurability, and pixel numbers are not comparable to commercially available off-the-shelf components (DMD or SLM). But even with limited pixel size, we are able to demonstrate high accuracy image classification and hyperspectral image processing. Also, compared to the free-space optic system, this on-chip metasystem has orders of magnitude smaller footprint, excellent mechanical robustness, is scalable with foundry processing, and low cost.

Our previous work [Ref. 25] demonstrated the design principle of the metasurface for integrated photonics and demonstrated a 4F (or 4-*f*) microsystem of metalens-metasurface mask-metalens. The design principle for spatial differentiation is well-known, and the small system can not implement any machine learning task. Theoretically, we can use the same integrated 4f framework for implementing CNN, however, the limited pixel size by nanofabrication in university has reached its limitation, and thus a single layer mask can not incorporate all the uncertainties needed for high accuracy image classification. And thus, we focus on a deep diffractive network designed multi-layer metasystem in this work. Its computing capacity, operation mechanism, and system scale are dramatically different from the previous work.

The tunability can be easily achieved in bulk components (DMD or SLM), however, it's a work in progress within the metasurface society. The programmable metasurface is achievable, with the

involvement of advanced phase change materials on the nanophotonic platform. However, the engineering and tuning of those materials are beyond the scope of current work. Combined with those electro-optic switches based on high refractive index contrast phase change materials, the presented metasystem is a powerful and energy-efficient platform for optical computing

Here is our correspondent revision in the manuscript:

Page1: **‘With subwavelength structure enabled directional diffraction and dispersion engineering, the light propagation through multi-layer metasurfaces achieves wavelength-selective image recognitions on a silicon photonic platform at telecommunication wavelength.’**

‘Enabled by the subwavelength structures, metasurfaces are capable of high spatial resolution phase control, photon momentum steering, and high-efficiency diffraction [1-3]. Designed dispersion and diffraction enable multi-layer metasystems for powerful optical analog signal processing [4-15]. The metasystems can perform mathematical operations of the impinging electromagnetic wave with subwavelength resolution [16-20]. Fourier transform method designed passive metasurface systems demonstrate real-time spatial differentiation and edge detection [21-29]. Beyond the deterministic functions, spatial information classifications are also demonstrated in free-space optical systems. Designed by deep neural networks, multi-layer free-space diffractive optic elements perform high-accuracy image classification and logic computing in millimeter and micrometer wavelength ranges [30-34]. With only one layer of phase or amplitude coded mask, the light flow in the free-space 4F system mimics the matrix calculation in convolutional neural networks, achieving high accuracy image classifications with computer-aid postprocessing [35-36]. A diffractive processing unit based on paired spatial light modulator-CMOS sensor arrays implements powerful deep learning tasks, with intermediate optoelectronic signal conversions for signal flow between the optical units [37]. In those optical systems, the pixel or cell size in the coded plane is well beyond wavelength, which sets fundamental limitations on the diffraction efficiency and spectral engineering capability.’

We also utilized the dispersion property of the proposed meta-system, and designed a spectrometer and spectral-spatial pattern classification metasystem, as shown in the added Figure 5.

Figure 5. On-the-fly hyperspectral image classifier. (a) The simulated in-plane light distribution with the input pattern “Y” at 1570nm is superimposed on the optical image of a fabricated device. **(b)** Measured signals on 6 channels for the same input as (a) (squares with error bars), compared to the simulated distribution at the output plane. **(c)** The confusion matrix of the experimental results. Blue and red letters indicate the incident wavelength of 1530nm and 1570nm, respectively.

Additional table S2:

Table SII: Comparison of neuron networks-based image classifiers

Neuron network	Convolution Neural Network [S7]	Convolution Neural Network [S8]	Diffraction Neural network (This work)
Programmed layer(s)	One amplitude-only layer of DMD	One phase-only layer of diffractive optic elements	Phase-only layers of metasurface
Reconfigurable	Yes	No	No
Postprocessing	Required	Required	Maximal only
Hyperspectral	No	Possible	Yes

Kernel size	16×208×208		16×32×32	450×2
Dataset	MNIST	CIFAR	CIFAR-10	MNIST
Accuracy	98%(s)	63%(s)	51% (e)	96% (s), 92%(e)

(s): numerical simulation result. (e): experimental measurements.’

2. The reported benchmark tests are very simple. The forward structure for the proposed deep diffractive neuron networks is the same as that in Ref. [25]. As the title claims, the purpose of the proposed integrated photonic metasystem is to conduct image classification tasks. However, compared to Ref. [25], the image classification cases demonstrated in this manuscript are much simpler. I suggest to demonstrate a more interesting image classification case to enhance the novelty of the manuscript.

Response: Thanks for the reviewer’s comments. Based on the integrated metasystem design principle, we numerically demonstrate image classification with MNIST handwritten digit database (Supplementary note S4), with an accuracy of 96%. Experimentally, the complexity of the system is limited by the number of ports (grating coupler array). The ports need to be arranged in the way that the connecting waveguides to metasurface have the same phase shift and the accumulative fabrication-related phase distortion can be less than 0.5π ., which is very difficult for 784 of such ports. We tried to fabricate a large array of input free-space image–chip coupler arrays, but the accumulative errors from the fabrication variation among the long connecting waveguide significantly reduce the accuracy. To the best of our knowledge, the demonstrated ONN on an integrated photonic platform can only process small data sets, such as 360 Vowel phonemes (4-class) [Nat. Photonics 11, 441 (2017)] and 4 letter classification (‘A’, ‘B’, ‘C’, ‘D’) [Nature 569, 208-214 (2019)].

Alternatively, we exhibit the power of the metasystem by demonstrating a hyperspectral image classifier (New Fig. 5), which is not possible in bulk systems. We also challenge the spectral resolution for such a system for WDMs. Here we show the details about spectrum recovery. The intensity distribution of the 11 outputs array is $I = TS$, where T is the transmission matrix and S is the input spectrum. We use a similar method from [51], to reconstruct the input spectrum from the outputs. The pseudo-inverse of the transmission matrix can be used to reconstruct the input spectrum by $S = T^{-1}I$. The transmission matrix can be calibrated by minimizing $\|I - TS\|^2$. The metasystem is designed in a similar backpropagation algorithm, and the metasystem is designed

to focus the light at the 11 outputs to maximize the output signal. After the system is fabricated, we tested the intensity of 11 outputs and calibrated the transmission matrix (Figure R1).

Figure R1. Recovered spectrum from the testing data.

In the revised manuscript, we added the new results about spectrometer and spectral-spatial pattern classification (on page 6): ‘The complicated diffraction and interference allow one-to-one correspondence between the spatial distributions of the light and the laser wavelength [51]. Combining both features, we design and experimentally demonstrate a two-wavelength pattern classification system (Fig. 5). An optical image of the designed metasystem is shown in Fig. 5a. The input grating couplers design is same as the one in Fig. 3. The metasystem is composed of 2-layer metasurfaces with 600 phase shifters per layer. The 6 output ports are correspondent to pattern “X”, “Y”, and “Z” at 2 input wavelengths of 1530nm and 1570nm. For the input pattern of “Y” at 1570nm, the simulated light distributions on the output plane (grey curves in Fig. 5b) are consistent with measured data points (solid squares in Fig. 5b). The measured confusion matrix (Fig.5c) indicates the hyperspectral pattern classification accuracy of 70%, with an insertion loss of 14.2dB.’

Page 6: ‘The spectral resolution of such a metasystem is limited by the number of output ports. The spectral resolution of 7nm can be achieved with 11 output ports.’

3. Recently, nonlinear activation function is common in the optical neural networks. But the proposed integrated photonic metasystem is still a linear one.

Response: Thanks for the reviewer’s note. Indeed, the nonlinear function can make the neural

network more powerful, but to the author's awareness, most of the ONN works require electronic implementation for activation functions. Nonlinear optical components are proposed for activation functions, with limited experimental demonstrations. For example, In Ref. [RR1], the authors only applied the nonlinearity through the computer after each layer of linear operation.

The neural network made with linear functions can also be used for some applications. For example, Ref. [25] achieved the image classification with a linear system. For our case, in the training process, we added the Softmax function as the nonlinear normalized function for the last layer, and the system seems to provide satisfactory accuracy in classifications. It is also possible to use some optical nonlinear materials to apply the nonlinearity after each layer of metasurfaces. Since it is a technical challenge currently faced by the whole metasurface society, involving such nanoscale compact nonlinear components into metasytem needs dedicated efforts, and is beyond the scope of the current paper.

[RR1] Shen, Yichen, et al. "Deep learning with coherent nanophotonic circuits." *Nature Photonics* 11.7 (2017): 441-446.

We added the discussion about the nonlinearity in the discussion part:

Page 7: 'The reconfigurability and nonlinear activation functions can be introduced into the metasytem platform via hybrid integration of active materials. For example, phase change materials with high refractive index contrast can fill those slots and provide sufficient phase tunability [52] for a fully reprogrammable metasytem. Certain active materials exhibit exceptionally high nonlinear responses (such as two-photon absorption-related free carrier absorption or absorption saturation) and are transparent at telecommunication wavelength ranges, which can be integrated into the diffractive networks as nano-scale activation functions with solution processing [53-54].

[52] Wang, N. et al. Focusing and defocusing switching of an indium selenide-silicon photonic metalens. *Opt. Lett.* 46, 4088 (2021).

[53] Wang, F. et al. Controlling Microring Resonator Extinction Ratio via Metal-Halide Perovskite Nonlinearity. *Adv. Opt. Mat.* 9, 2100783 (2021).

[54] Wang, F. et al. Light Emission from Self-Assembled and Laser-Crystallized Chalcogenide Metasurface. *Adv. Opt. Mat.* 8, 9101236 (2020).'

Some minor issues:

4. In Fig. 2b, I notice that some values of the normalized intensity is above 1 according to the error bar.

Response: Thanks for the reviewer’s careful check. We re-visited the original data and corrected its scale. Here is the updated Fig. 3:

5. Some grammatical mistakes and typos are found, and hence the language quality needs to be improved.

Response: Thanks for the reviewer’s note. We have double-checked our revised manuscript for the typos and grammatical errors.

Reviewer 2:

By using the deep diffractive neuron networks trained silicon metasystem, the authors demonstrate the function of image recognition. The metasystem can do high-throughput vector-by-matrix multiplications, enabled by the passive subwavelength phase shifters. A 15-pixel spatial pattern classifier can reach near 90% accuracy. The metasystem can also potentially improve the data

processing capability of the CMOS compatible photonic integrated circuits. The miniaturized image classifiers have great applications potentials in optical communication, autonomous vehicles, and healthcare. This is a very interesting work; the experimental results agree with the numerical simulations. Therefore, I would like to support its publication after revisions.

Comments:

1. The optical efficiency of the metasystem should be discussed in more details.

Response: Thanks for the reviewer’s inquiry. Insertion loss is one of the critical parameters in the integrated photonic component/system’s performance matrix. In those reported integrated photonic metasystems, the total insertion loss varies between components. The value is around 15 dB. This value is added into Table SI for comparison among integrated photonic devices.

We also added the correspondent insertion loss for each metasystem in the main text:

Page 5: ‘Numerical simulation shows the insertion loss in the metasystem classifier is 9.3dB.’

Page 6: ‘The measured insertion losses for such three-layer system are 13.1dB, 16.8dB and 18.9dB for the wavelength at 1490nm, 1530nm and 1570nm, respectively.’

‘the hyperspectral pattern classification accuracy is about 70%, with an insertion loss of 14.2dB.’

Figure 1a, the phase shifters have low image resolutions, I would suggest using less shifters to show the concept. Figure 1c and 1d, it is not easy for the audience to understand. The basic configuration of incident plane of light should be depicted.

Response: Thanks for the reviewer’s suggestions. We updated figure 1a with fewer shifters for better illustration of the device schematics.

2. Figure 2d, for the non-expert, it is difficult to understand the working principle. For example, the SMF is used to collect the light, but people can not see where is the excitation. It is better to re-plot Figure 2a and 2d. The structures on the DUT are not clear as well.

Response: Thanks for the reviewer’s advice. We are sorry about the confusion. We have enlarged the DUT (optical microscope image of the device placed in perspective) in figure 2a and replaced figure 2d with a higher resolution SEM image. The positions of the nanostructures (as enlarged SEMs in c-d) are identified in the optical image embedded in Fig. 2a.

Here is the revised Fig. 2:

3. In the discussion part, the authors mention that “compared to the 2D metasurface, the number of the elements is significantly reduced”. This is true, however, I would expect the efficiency and the quality image recognition by using a 2D metasurface is even better. Please explain more about this.

Response: Thanks for the reviewer’s comments. We add the discussion of limitation and correspondent mitigation strategy in the discussion section. 1D metasurface format set limitations of the number of cells per layer but expand the depth of the metasystem. For the fixed number of

cells (untunable neuron), the classification accuracy is more sensitive to the depth of the metasystem than the number of cells per layer (Fig. S7a).

Figure S7. Scalability of the integrated diffractive optical network for MNIST. (a) Simulated accuracy versus the same total number of the weight elements, with a different number of layers (N). The inter-layer distance is 100 μm. (b) Accuracy versus layer number with inter-layer distance (D) of 50, 100, 250, and 500 μm. The number of phase shifters per layer is 1000.

Also, we want to emphasize that 2D metasurface for machine learning has *never* been experimentally demonstrated in telecommunication or infrared wavelengths, even for a single layer metasurface-based convolutional neural network. We believe that the strategy of consideration of fabrication and alignment variation during the training section can help facilitate such systems in a multi-layer 2D metasurface.

We have added additional discussions on the pros and cons of 1D metasurface:

Page 6-7: ‘Compared to the 2D metasystem in free space, the metasurface on the integrated platform is limited to a smaller number of cells and out of plane-in plane couplers, with the advantages of lower insertion loss and feasible fabrications for multi-layer structures. With the same total cell number, classification accuracy is more sensitive to the depth of metasystems than the size of each layer (Fig. S7a). Currently, the fabrication limited metasurface cell number is 10⁴, which is sufficient for the standard testing databases with propagation matrix compression (Supplementary Note S2.2). We numerically explored the 1D metasystem’s computing capability by designing one for a Modified National Institute of Standards and Technology (MNIST) handwritten digit database with 784-pixel inputs (Supplementary Note S4). 3 Epochs bring a metasystem’s accuracy to be 96% (Fig. S6). Currently, the main technical challenge is the layout design of a large number of I/O ports on an integrated photonic platform with tolerable phase distortions from nanofabrication. Theoretically, a 2D metasurface with the subwavelength cell

owns significant computing capabilities. However, experimental implementation of such a system for machine learning has never been reported in telecommunication wavelength or infrared, but feasible if the fabrication or alignment errors are considered in the training process (Supplementary Note S2.1 and S2.5). Commercially available components (DMD or diffractive optical elements) have a typical cell number of 10^4 - 10^6 . Single layer component has been utilized for high-accuracy image classifications [35-36]. The integrated photonic platform can eliminate out-of-plane light diffraction, and thus result in orders of magnitude lower insertion loss compared to free-space optical systems. ’

4. Many important literatures in the field of metasurfaces are not included in the references.

Response: Thanks for reviewer’s note. We added literatures in the field of metasurfaces in the references. In the introduction part, we provide a more complete citations of the milestone publications on metasurface, as well as include the description of metasurface enabled holography technology.

Page 1: ‘Enabled by the subwavelength structures, metasurfaces are capable of high spatial resolution phase control, photon momentum steering, and high-efficiency diffraction [1-3]. Designed dispersion and diffraction enable multi-layer metasystems for powerful optical analog signal processing [4-15]. The metasystems can perform mathematical operations of the impinging electromagnetic wave with subwavelength resolution [16-20].’

Additional citations:

[1] Ni, X., Kildishev, A. V. & Shalaev, V. M. Metasurface holograms for visible light. *Nat Commun* 4, 2807 (2013).

[2] Zheng, G. et al. Metasurface holograms reaching 80% efficiency. *Nature Nanotech* 10, 308–312 (2015).

[3] Ren, H. et al. Complex-amplitude metasurface-based orbital angular momentum holography in momentum space. *Nat. Nanotechnol.* 15, 948–955 (2020).

[6] Lee, G.-Y. et al. Metasurface eyepiece for augmented reality. *Nat Commun* 9, 4562 (2018).

[14] Lin, D., Fan, P., Hasman, E. & Brongersma, M. L. Dielectric gradient metasurface optical elements. *Science* 345, 298–302 (2014).

[15] Jahani, S. & Jacob, Z. All-dielectric metamaterials. *Nature Nanotech* 11, 23–36 (2016).

Reviewer 3:

The manuscript under review deals with the realization of an optical analog signal processor to enable image classification in the infrared range. The diffractive nature of the proposed platform comprising of cascaded 1D metasurfaces somehow mimics the functionality of a trainable artificial neural network. Such a passive metasystem provides acceptable performance measures including classification accuracy, throughput, and footprint. Overall, this work is a timely contribution to the on-demand topic of biologically inspired optical computing. The manuscript is organized, and the results look sound. I believe that this manuscript can stimulate further research lines in visual computing, image analysis, and feature detection. Said that, I think that the work is important enough to be published in *Nat. Commun.* However, there are several major concerns/suggestions that need to be addressed before I can recommend its publication.

Response: First, the authors sincerely appreciate the reviewer's time and efforts for such a careful check and provided extended suggestions and comments on our work.

1. There exist two major concerns with optical neural networks (ONN): i) lack of nonlinearity (mimicking the nonlinear activation function of a neuron) which limits the functionality of ONN to a simple linear transformation of the incident light, ii) absence of tunability which hinders the online training of neurons. This hinders the realization of biologically inspired computing paradigms. Can the authors comment on how their platform can address these issues?

Response: We are sorry about the confusion. ONN is a very interesting topic with great potentials for hardware accelerators. However, the purpose of this work is not to demonstrate an ONN, but the presentation of a metasurface-based architecture for implementing diffractive optical networks (To avoid confusion, the term ONN is not mentioned throughout the manuscript). The tunability and activation functions are critical components for ONN, but beyond the scope of the current work. We want to emphasize that the novelty of the presented work. This work is the first demonstration of a passive metasurface system (metasystem) for machine learning tasks at telecommunication wavelengths (on both free-space and on-chip platforms).

In the supplementary file S1, we compare the performance matrix of the metasystem to the other integrated photonic subsystems for basic matrix-vector multiplications. Indeed, the nonlinear function can make the neural network more powerful, the neural network made with linear functions can also be used for some applications. For example, Ref. [25] achieved the image

classification with a linear system. In Ref. [R1], the authors only applied the nonlinearity through the computer after each layer of linear operation. For our case, in the training process, we added the Softmax function as the nonlinear normalized function for the last layer, and we did achieve pattern classification with a purely linear system in the experiment. It is also possible to use some optical nonlinear materials to apply the nonlinearity after each layer of metasurfaces. The nonlinear activation function is considered in the estimation of operational power in table SI.

The active devices are indeed powerful, however, the passive devices can be used for a few application-specific cases with the advantages of easier fabrication (low cost) and zero power consumption for maintaining the phase tuning. Also, we can design the on-chip metalens for Fourier transform as we have shown in [36] and connect the photodetector output with the electronic part, to achieve tunability.

[R2] Shen, Yichen, et al. "Deep learning with coherent nanophotonic circuits." *Nature Photonics* 11.7 (2017): 441-446.

[R3]: Chang, J., Sitzmann, V., Dun, X., Heidrich, W. & Wetzstein, G. Hybrid optical–electronic convolutional neural networks with optimized diffractive optics for image classification. *Sci. Rep.* 8, 12324 (2018).

Additional discussion on page 7: ‘The reconfigurability and nonlinear activation functions can be introduced into the metasystem platform via hybrid integration of active materials. For example, phase change materials with high refractive index contrast can fill those slots and provide sufficient phase tunability [52] for a fully reprogrammable metasystem. Certain active materials exhibit exceptionally high nonlinear responses (such as two-photon absorption-related free carrier absorption or absorption saturation) and are transparent at telecommunication wavelength ranges, which can be integrated into the diffractive networks as nano-scale activation functions with solution processing [53-54].

[52] Wang, N. et al. Focusing and defocusing switching of an indium selenide-silicon photonic metalens. *Opt. Lett.* 46, 4088 (2021).

[53] Wang, F. et al. Controlling Microring Resonator Extinction Ratio via Metal-Halide Perovskite Nonlinearity. *Adv. Opt. Mat.* 9, 2100783 (2021).

[54] Wang, F. et al. Light Emission from Self-Assembled and Laser-Crystallized Chalcogenide Metasurface. *Adv. Opt. Mat.* 8, 9101236 (2020).’

Also, we updated table SI and add additional comparison among image classifiers as table SII.

Page S-1: ‘Table SI: Integrated photonic frameworks for VMM.

Method for matrix operation	Multi-wavelength modulation and summation [S2-S4]	Singular value decomposition [S5]	Diffraction equation (This work)
Device architecture	MRR weight banks [S2, S4] and directional couplers [S3]	MZI[S5]	Cascaded metasurfaces
Signal processing	Optoelectronic	Optoelectronic	All-optical
Weight matrix	16×16 [S3]	4×4 [S5]	450×2
Footprint (mm ²)	16 [S3]	0.75[S5]	0.135
Throughput (Tb/s)	11 [S2]	1 [S5]	5
Insertion loss	27dB [S3]	Not reported	15 dB (average)
Operational power	17fJ per MAC	1pJ per FLOP	10 ⁻⁵ fJ per FLOP

Table SII: Comparison of neuron networks-based image classifiers

Neuron network	Convolution Neural Network [S7]		Convolution Neural Network [S8]	Diffraction Neural network (This work)
Programmed layer(s)	One amplitude-only layer of DMD		One phase-only layer of diffractive optic elements	Phase-only layers of metasurface
Reconfigurable	Yes		No	No
Postprocessing	Required		Required	Maximal only
Hyperspectral	No		Possible	Yes
Kernel size	16×208×208		16×32×32	450×2
Dataset	MNIST	CIFAR	CIFAR-10	MNIST
Accuracy	98%(s)	63%(s)	51% (e)	96% (s), 92%(e)

(s): numerical simulation result. (e): experimental measurements.’

2. The design principle of the metasurfaces is based on the arrangement of length-variant slots (Fig. 1c). However, Fig. 3a clearly shows that all slots have the same lengths. Can the authors comment on this?

Response: We are sorry about the confusion. In Fig. 3a, different colors of metasurface cell stand for different phase shifts. In the revised manuscript, we replace the color bar with an optical image of the correspondent device and superimpose the device image onto the simulated optical field distribution.

Updated Fig. 3:

3. Given that in one direction the length of building blocks is up to 3 μm, can we still call “subwavelength phase shifters”? This brings a more fundamental question to the stage; can we call the proposed structure a “metasystem”?

Response: We are sorry about the confusion from the distorted scale bars. The width and period of the slot are 100nm and 500nm respectively. The slot lengths vary between 0-3.2μm. We think we can still call the proposed structure a “metasystem” since the proposed multi-layer structure

can modify the phase or the amplitude of the transmitted light in the subwavelength resolution.

We have updated all the scale bars in the figures.

4. I assume that the authors adopted the same configuration in [Nature Commun. 10.1 (2019): 1-7], in which the periodicity is considered 500 nm. Considering Fig. 3a, the length of the metasurface (in the x-direction) is ~ 130 μm which means that ~ 260 meta-atoms are effectively in charge of signal processing. This is way lower than the 450 reported in the main text and Table S1. Can the authors comment on this discrepancy? If this is the case, then almost half of the slots are idle and have no contribution to the calculated 2×450 weight matrix.

Response: We are sorry about the confusion. The aspect ratio in Fig. 3a is a little distorted when we adjusted the figure. In Fig. 3a, and the number of neurons is indeed 450 for each layer. We updated Fig. 3a and the aspect ratio in the updated figure is 1:1 and correct the scale bar. Also, we add the missing information of lattice constant in the main text.

Page 3: ‘With a fixed slot width of 100nm and lattice constant of 500nm,...’

The scale bar in Fig. 3a is updated as well.

5. The authors claimed that their classifier is robust against the fabrication imperfections. While this is inherently afforded by large-scale metasurfaces, it needs, at least in terms of FDTD simulations, to be justified. Confusion matrices for different scenarios can be provided.

Response: Thanks for the reviewer’s comments. We have added this additional simulation result in the supporting information section. Here is our correspondent revision in the supporting material:

Page S-7: ‘**S2.5 Interval range for phase noise robustness**

The interval incorporates the fabrication offsets. During the ebeam lithography and etching process, random variations of slot width and length < 10 nm are expected, and more variation can be expected for inter-layer distance ($100\mu\text{m} \pm 10$ nm). Besides the error caused by fabrication, the experiment setup and measurement can also cause the error. A larger interval provides better robustness against error but also decreases the classification accuracy. The interval $[0, 0.5\pi)$ is selected for balancing the design robustness and accuracy.

Figure S4. Comparison of classification accuracy with and without considering the phase noise. (a) and (b) The FDTD simulated confusion matrices for pattern classification systems with and without the phase noise added during the training step.’

6. Given 90-fs-long pulses, how the authors reached a throughput of 10^{15} b/s? Above that, considering the long-distance dispersive integrated platform, what is the delay (and corresponding speed) associated with the optical processing? Did the author consider this factor when calculating 1000 Tb/s throughput? Can the authors comment on the speed of the photodetector?

Response: Thanks for the reviewer’s comments. The throughput value in the original table SI was wrong. The throughput should be about $5 * 10^{12} b/s$. We only considered the device's limited operation speed but have not taken into consideration of the photodetector limitations. For on-chip photodetector, maximum bandwidth up to 265GHz is recently reported (Lischke et al, Nat. Photonics 15, 925-931, 2021), which is sufficient for the derived value here. The equation (R1) describes the optical device limited operation frequency:

$$f_{DPATH} = \frac{1}{L} = \frac{1}{S * \frac{n}{c}} \quad (R1)$$

where L is the latency, S is the distance of the optical data path, n is the refractive index, c is the speed of light. Throughput of the system is calculated by:

$$Throughput = N * f_{DPATH} \quad (R2)$$

where N is the dimension of the input data. After such calculation, the system could reach a throughput of $5 * 10^{12} b/s$.

In addition, we double-checked the power consumption calculations. In the revised manuscript, we have added these annotations for table S1 in the correspondent section.

Page S-1: ‘Calculations of throughput and power consumption in the table S1 are detailed below.

Throughput: The equation (S1) described the optical data path limited operation frequency:

$$f_{DPATH} = \frac{1}{\tau} = \frac{1}{S/v} \quad (S - 1)$$

Where τ is the latency, S is the distance of the optical data path, v is the speed of light propagation in the media. Throughput of the system is calculated by:

$$Throughput = n_i f_{DPATH} \quad (S - 2)$$

where n_i is the dimension of the input data. After such calculation, the system could reach a throughput of $5 * 10^{12} b/s$. The number of operations per second of the system is:

$$FLOPS = mNNf_{DPATH}, \quad (S - 3)$$

where N is the neuron number per layer, and m is the layer number [S6].

Operational power: The power consumption of the metasystem is the summation of the power required for propagation and the optical power required to support an optical nonlinearity that could be potential implementations of future devices. If we assume a saturable absorber threshold of $p \cong 1MW/cm^2$ (e.g. graphene) and an area of a neuron $A = 1\mu m^2$, the total power needed for nonlinearity is estimated to be $P = p \times A \times N = N(mW)$. For the proposed two-layer system, the power consumption is:

$$\frac{P + Loss}{FLOPS} = \frac{450 \times 1.9 \times 10^{-3}}{2 \times 450 \times 450 \times 5 \times 10^{12}} \cong 4.2 \times 10^{-19} J \text{ per FLOP}. \quad (S - 4)$$

Where $Loss$ is the insertion loss of the system.'

7. The 1D nature of the proposed platform (compared to free-space metasurfaces) highly limits the classification of high-resolution images. The authors demonstrated 15-bit inputs. Can the authors discuss this limitation and compare it with 2D metasystems?

Response: Thanks for the reviewer's comments. We add the discussion of limitation and correspondent mitigation strategy in the discussion section. 1D metasurface format set limitations on the number of cells per layer but expand the depth of the metasystem. For the fixed number of cells (untunable neuron), the classification accuracy is more sensitive to the depth of the metasystem than the number of cells per layer (Fig. S7a).

Figure S7. Scalability of the integrated diffractive optical network for MNIST. (a) Simulated accuracy versus the same total number of the weight elements, with different number of layers (N). The inter-layer distance is 100 μm . **(b)** Accuracy versus layer number with inter-layer distance (D) of 50, 100, 250, and 500 μm . The number of phase shifters per layer is 1000.

Also, we want to emphasize that 2D metasurface for machine learning has *never* been experimentally demonstrated in telecommunication or infrared wavelengths, even for a single layer metasurface-based convolutional neural network. We believe that the strategy of consideration of fabrication and alignment variation during the training section can help facilitate such systems in a multi-layer 2D metasurface.

We have added additional discussions on the pros and cons of 1D metasurface:

Page 6-7: ‘Compared to the 2D metasytem in free space, the metasurface on the integrated platform is limited to a smaller number of cells and out of plane-in plane couplers, with the advantages of lower insertion loss and feasible fabrications for multi-layer structures. With the same total cell number, classification accuracy is more sensitive to the depth of metasytems than the size of each layer (Fig. S7a). Currently, the fabrication limited metasurface cell number is 10^4 , which is sufficient for the standard testing databases with propagation matrix compression (Supplementary Note S2.2). We numerically explored the 1D metasytem’s computing capability by designing one for a Modified National Institute of Standards and Technology (MNIST) handwritten digit database with 784-pixel inputs (Supplementary Note S4). 3 Epochs bring a metasytem’s accuracy to be 96% (Fig. S6). Currently, the main technical challenge is the layout design of a large number of I/O ports on an integrated photonic platform with tolerable phase distortions from nanofabrication. Theoretically, a 2D metasurface with the subwavelength cell owns significant computing capabilities. However, experimental implementation of such a system for machine learning has never been reported in telecommunication wavelength or infrared, but

feasible if the fabrication or alignment errors are considered in the training process (Supplementary Note S2.1 and S2.5). Commercially available components (DMD or diffractive optical elements) have a typical cell number of 10^4 - 10^6 . Single layer component has been utilized for high-accuracy image classifications [35-36]. The integrated photonic platform can eliminate out-of-plane light diffraction, and thus result in orders of magnitude lower insertion loss compared to free-space optical systems.'

8. I could not find Ref. [3] related to the optical signal processing as it mainly discusses the nanophotonics design using deep learning approaches. Also, the realization of mathematical operations using metamaterials is not covered in Refs. [14, 15] as mentioned in the introduction section. I found the following works more related: [Nat. Commun. 8, 15391 (2017)], [Nano Lett. 15.1 (2015): 791-797], and [Optics Lett. 40.22 (2015): 5239-5242].

Response: Thanks for the reviewer's comments. We have replaced the references and added those citations in our updated manuscript.

We added the citations in the reference part:

[27] Zhu, T. et al. Plasmonic computing of spatial differentiation. Nat Commun 8, 15391 (2017).

[28] Pors, A., Nielsen, M. G. & Bozhevolnyi, S. I. Analog computing using reflective plasmonic metasurfaces. arXiv:1609.04672 [physics] (2016).

[29] AbdollahRamezani, S., Arik, K., Khavasi, A. & Kavehvas, Z. Analog computing using graphene-based metalines. Opt. Lett. 40, 5239 (2015).

9. The authors motivated their work by referring to the possible challenge of interlayer misalignment of multilayered metasurfaces. However, several works (e.g., [Optica 7.1 (2020): 77-84] and [Nano letters 18.12 (2018): 7529-7537]) experimentally demonstrated that such an issue can be mitigated through highly accurate bonding processes or integration of vertical heterostructures. Can the authors provide more motives/rationales for the proposed architecture?

Response: Thanks for the review for bringing up those impressive works to the authors' awareness. Both of those works rely on lithography-aligned precision between metasurface layers and require increasing times of alignment with the number of metasurface layers. In our device architecture, one-step alignment is needed for any number of metasurface layers, which eliminates the multi-layer alignment error and is simple to implement.

Without considering the alignment issue, cascaded two-dimensional metasurface layers in free space are perceived to exhibit powerful computing capabilities [Backer Opt. Exp. 27, 30308. Ref. 19]. Here we found that the propagation matrix in 2D metasurface can be compressed to 1D (supporting information S2.2). Compared to 2D systems [19], the 1D metasystem significantly reduced metasurface cells (from $O(N^2)$ to $O(N)$) with satisfactory classification accuracy.

We have included the two suggested papers in the manuscript and revised the related discussion in the introduction.

Page 2: ‘Metasurface-based multi-layer systems, named metasystem, expand the functionality of metasurface in the out-of-plane dimension [38-40]. Lithographically assisted alignment and bonding between metasurface layers are required for providing sufficient precision and robustness in functional metasystems [39-40]. The integrated photonics platform provides such alignment with one-step lithographically defined multiple metasurface layers.’

[38] McClung, A., et al. Snapshot spectral imaging with parallel metasystems. *Sci. Adv.* 6, eabc7646 (2020).

[39] Mansouree, M., et al. Multifunctional 2.5D metastructures enabled by adjoint optimization, *Optica* 7, 77-84 (2020).

[40] Zhou, Y., et al. Multilayer noninteracting dielectric metasurfaces for multiwavelength metaoptics. *Nano letters* 18, 7529-7537 (2018).’

10. Can the authors comment on why the phase distortion with oblique incidence is negligible?

Response: Thanks for the reviewer’s comment. Fig. 1d shows the calculated phase of transmission coefficient versus incident angle. Less than $0.01(\times 2\pi)$ phase deviation (blue curve in Fig. 1d) is observed between 0 and 40 degrees of incident angle. We add the quantitative details in the main text.

Also, we did an additional investigation on the deviation’s dependence on slot length. From the 3D FDTD simulation (according to the reviewer’s suggestion in question 12), the low phase distortion persists at varying slot lengths. We add the new simulation result in the supporting information.

Page 3: ‘Fig. 1d shows the angle-dependent complex transmission coefficient. The amplitude of the transmission reduces to half as the incident angle increases from 0° to 28° , with phase distortion less than 0.01 (in the unit of 2π rad). The results in Fig. 1d are insensitive to the slot length (Fig.

S3).’

Page S-6: ‘

Figure S3. Incident angle-dependent transmission and phase deviation of the slots with a fixed width of 100 nm and varying lengths (0.2-3.6 μ m). The result is obtained by 3D FDTD simulation.’

11. “Both amplitude modulation the phase shift of the transmitted wave can be programmed by adjusting the width and length of the subwavelength slots (Fig. 1c-d) [36].”, While Fig. 1c shows the variation of amplitude/phase as a function of the length of the slot, Fig. 1d represents this measure versus the angle of incidence. It would be better to provide such measures as a function of both width/length in the Supplementary and revise the text accordingly.

Response: Thanks for the reviewer’s comment. Here are the simulated transmission and phase shift versus slot lengths and widths. Those results are included in our previous publication, as shown in figure 1 in [48]. Fig. 1c shows the design principle for the ‘phase-only’ metasurface, where only the design with the unit transmission is used here for minimizing insertion loss.

Here we added the simulation results in the supplementary part and revised the main text accordingly.

Page 3: ‘Both amplitude modulation **and** the phase shift of the transmitted wave can be programmed by adjusting the width and length of the subwavelength slots [48] (Fig. S2).’

Page S-5: ‘**S2.3 Metasurface cell design**

The transmission and the phase shift of the transmitted light can be modified by changing the geometric parameters of the slots as shown in Fig. S2. Also, Fig. S3 shows that the phase shift is not sensitive to the incident angle. Phase deviation less than $0.02 \times 2\pi$ is found at the incident angle

of 40°.

Figure S2. Fully programmable complex transmission coefficient of the metasurface cell. The simulated (a) transmission and (b) phase shift versus slot length and width.'

12. The width of the meta-atom in Fig. 1C and width/length of the meta-atom in Fig. 1D should be mentioned in the caption. Can the authors get similar curves for different slot lengths by changing the angle of incidence?

Response: Thanks for the reviewer's comment. The width of the slots throughout the paper is fixed at 100nm, and the length of the slot in Fig. 1d is 2.5μm. That information has been added to the figure and figure caption. Additional full-field simulation is performed to better understand the angle-dependent complex transmission for different slot lengths and added as a few Figure S3. Like the result in Figure 1d, the phase shift is not sensitive to the incident angle at the slot length range of 0.2-3.6μm.

The correspondent manuscript revision is the same as the one for question 10.

13. While the first layer of the metasurface accepts a small angle of incidence, this is not the case for layer 2 (and 3). It means that the amplitude variation across the intermediate layers is not negligible. This highly affects the vector-matrix multiplication formula governing the operation of DNN. Can the authors verify this?

Response: Yes, indeed it can affect the vector-matrix multiplication formula. We are sorry that we failed to mention A Gaussian-like modulation was added to modulate the amplitude of the outputs for each layer. We added $U(\Delta y) \propto e^{-[\frac{\pi \Delta y \sigma}{\lambda a}]^2}$ to modulate the outputs of each layer, where Δy is the relative distance along y-direction between pixels, a is the spacing along the x-direction. σ was

chosen to be $0.45\mu\text{m}$ and $0.08\mu\text{m}$ for the input layer and other layers. With this modulation, the intensity of the light with the large angle will reduce, and the python simulations and the FDTD simulations matched well.

This information is added into the manuscript:

Page 4 ‘Considering the angle-dependent transmission amplitude (Fig. 1d), an additional factor of $U(\Delta y) \propto e^{-[\frac{\pi\Delta y\sigma}{\lambda a}]^2}$ is superimposed onto the outputs of each layer, where Δy is the relative distance along the y -direction, a is the spacing along the x -direction. σ is $0.45\mu\text{m}$ for the first layer and $0.08\mu\text{m}$ for the subsequent layers, obtained by fitting the model to the numerical simulation results.’

14. “The input grating couplers are one-dimensional (1D) gradient metasurface and about $10\mu\text{m}$ wide in x direction (Fig. 2d).”, to my eye, the width of grating is $> 13\mu\text{m}$. Can the authors report the exact values? Also, adding an optical image of a single device with enlarged SEM images of building blocks can be more illustrative.

Response: Sorry about the confusion. The scale bars were wrong. The width of the grating (in x direction) is $20\mu\text{m}$. We also replace the original SEMs with higher resolution ones (updated Fig. 2c-d below).

The optical image of a single device is also embedded in Figure 2a. Additional notations identifying the position of magnifying SEMs (Fig. 2c-d) are superimposed onto this image.

Here is our correspondent revision in the manuscript:

On page 5: ‘). The scanning electron microscope (SEM) images show the detailed nanostructures of the grating coupler array (Fig. 2c) and the pre-trained metasurface (Fig. 2d) on DUT.’

We also updated Fig. 2:

15. Figure 3b has different scales for simulation and measured data. If the measured optical intensities are normalized why the error bar exceeds 1? Can the authors comment on the normalization strategy?

Response: Thanks for the reviewer's careful check. We have corrected the scale for experimental data in figure 3b. Here is the revised Fig. 3:

16. “Each phase shifter in the hidden layer is represented by two subwavelength slots ...”. Based on Fig. 1, the design methodology is based on formation of a single slot. Where does this discrepancy come from?

Response: Sorry about the confusion. The complex transmission coefficients in Fig. 1c-d are not calculated from a single slot, but a periodic array of the identical design (periodic boundary applied in the numerical simulation for obtaining those results). Based on the array calculation result, we found a noticeable discrepancy to the single slot design in numerical simulations, and thus we increase the number of slots in each cell to reduce the discrepancy.

Correspondent revision in the manuscript to clarify this:

Page 3: ‘Distinguished from our prior demonstration of gradient metasurface-based mathematical operators, large phase contrasts between neighboring cells are required in the metasurfaces for machine learning tasks. As the transmission coefficient of each metasurface design is numerically calculated from a periodic array, single-slot implementation of each phase shifter in gradient metasurface design results in unexpected discrepancies, and thus two subwavelength slots are employed here for representing one phase shifter in the designed network [48] (Inset of Fig. 1d).’

17. To improve the system’s robustness against nanofabrication, random phase noise in the interval of $[0,0.5\pi)$ is added during the training process. What is the reason behind selecting this interval?

Response: Thanks for the reviewer’s comment. The intervals indicate the random fabrication offsets or free space measurement phase fluctuations (detailed in figure S1a). During the ebeam lithography and etching process, random variations of slot width and length <10 nm are expected, and more variation can be expected for inter-layer distance ($100\mu\text{m} \pm 10$ nm). Given the accumulative random offset, we tried different intervals, and the one $[0,0.5\pi)$ works the best. A larger interval provides better robustness against error, but we also observed that it also decreases the classification accuracy. The interval is selected with the compromise between classification accuracy and the large interval.

Page S-7: ‘**S2.5 Interval range for phase noise robustness**

The interval incorporates the fabrication offsets. During the ebeam lithography and etching process, random variations of slot width and length <10 nm are expected, and more variation can be expected for inter-layer distance ($100\mu\text{m} \pm 10$ nm). Besides the error caused by fabrication, the experiment setup and measurement can also cause the error. A larger interval provides better robustness against error but also decreases the classification accuracy. The interval $[0,0.5\pi)$ is selected for balancing the design robustness and accuracy.

Figure S4. Comparison of classification accuracy with and without considering the phase noise. (a) and (b) The FDTD simulated confusion matrices for pattern classification systems with and without the phase noise added during the training step.’

18. The bandwidth of the femtosecond laser in the main text is given “near 20 nm” while in the caption of Fig. 3 is “over 30 nm” and in the Methods is “around 50 nm”. Consistency is important throughout the manuscript.

Response: Sorry about the confusion. The bandwidth of the femtosecond laser is 30 nm centered

at 1551.6 nm.

Here is our correspondent revision in the manuscript:

Page 5: ‘The broadband operation is critical for ensuring high classification accuracy for single-shot ultrafast pulsed inputs. Under 90 femtosecond pulsed light (centered at 1551.6 nm with a bandwidth of 30 nm), ...’

Page 8: ‘a femtosecond laser centered at 1551.6 nm with a duration less than 90 fs and spectral bandwidth around 30 nm (Calmar laser CFL-10CFF) is used to replace the continuous wave light source.’

19. Why is the interlayer distance chosen 100 μm ? Why is 30 μm spacing considered in the output?

Response: The interlayer distance was set to be 100 μm by compromising the footprint, loss, and recognition accuracy. Smaller inter-layer distance results in lower accuracy (Fig. S7b), and larger distance increases insertion loss.

The spacing of the outputs for the pattern recognition systems is 100 μm on the output plane. In the WDM system, the spacing is 30 μm . The spacing between the outputs can be changed, but close-spaced output ports design result in lower accuracy.

Here is our correspondent revision in the manuscript:

Page 4: ‘The inter-layer distances are selected to be 100 μm , balancing the insertion loss and classification accuracy.’

20. According to the confusion matrix in Figs. 3d, the classification accuracy for CW is ~ 94.4%.

Response: Thanks for the reviewer’s comment. We double-checked the confusion matrix, the accuracy for Fig. 3d is calculated as: $\frac{14+15+15}{16*3} \cong 92\%$.

21. In my view, Table S1 is not accurate. The signal format for Refs. [S2, S3] are temporal-spectral, the signals are coherent, and their architectures are based on MZIs and directional couplers, respectively. I recommend the authors revise the table according to the reported data provided in those papers.

Response: Thanks for the reviewer’s comment. We fixed the table SI after studying those references again.

Page S-1: ‘Vector-by-matrix multiplication (VMM) is one of the fundamental operations in the accelerator hardware [S1]. Table S1 compares the VMM power efficiency, throughput, and footprint of integrated photonic circuits based on Mach-Zehnder interferometer (MZI), microring resonators (MRRs), and metasystem (this work).

Table SI: Integrated photonic frameworks for VMM.

Method for matrix operation	Multi-wavelength modulation and summation [S2-S4]	Singular value decomposition [S5]	Diffraction equation (This work)
Device architecture	MRR weight banks [S2, S4] and directional couplers [S3]	MZI[S5]	Cascaded metasurfaces
Signal processing	Optoelectronic	Optoelectronic	All-optical
Weight matrix	16×16 [S3]	4×4 [S5]	450×2
Footprint (mm ²)	16 [S3]	0.75[S5]	0.135
Throughput (Tb/s)	11 [S2]	1 [S5]	5
Insertion loss	27dB [S3]	Not reported	15 dB (average)
Operational power	17fJ per MAC	1pJ per FLOP	10 ⁻⁵ fJ per FLOP

- Also, the footprint is given 1 mm² in the abstract while reported 0.045 mm² in the table. Moreover, the number of phase shifters in the abstract is 10³ while in the table is 2×450.

Response: Sorry for the confusion. The size reported in the abstract includes all the input and output ports (waveguides and grating couplers). Those parts are supporting interfaces for light coupling between the metasystem and fibers. Their sizes are limited by the single-mode fiber, but not the metasystems. The key component of the metasystem is 0.045 mm² per layer. To make a fair comparison, we updated the table with footprint excluding the waveguides and couplers. Also, we updated the metasystem size in the abstract.

‘The metasystem implements high-throughput vector-by-matrix multiplications, enabled by near 10³ nanoscale phase shifters as weight elements within 0.135 mm² footprints.’

- Finally, WDM stands for wavelength division multiplexing that needs to be corrected!

Response: Thanks for the reviewer's comment. In the original manuscript, 'microring resonator (WDM)' was meant for indicating the term 'WDM' used in the table. This problem is resolved in the revised supplementary note S1.22. The authors did not discuss the optical performance of their architecture. This is an important performance measure (specifically for scalability) that needs to be compared with other PIC architectures.

Response: Thanks for the reviewer's suggestion and reminder. Indeed, we compared the footprint, throughput, and power of the proposed device architecture to WDM and MZI based systems. Beyond those performance matrices, the insertion loss is also an important figure of merit. The total insertion loss varies between components. The value is around 15 dB. This value is added into Table SI for comparison among integrated photonic devices.

We also added the correspondent insertion loss for each metasystem in the main text:

Page 5: 'Numerical simulation shows the insertion loss in the metasystem classifier is 9.3dB.'

Page 6: 'The measured insertion losses for such three-layer system are 13.1dB, 16.8dB and 18.9dB for the wavelength at 1490nm, 1530nm and 1570nm, respectively.'

'the hyperspectral pattern classification accuracy is about 70%, with an insertion loss of 14.2dB.'

23. Why are the output ports in Fig. S2 100 μm away from each other?

Response: Sorry about the confusion. The spacings between the outputs for all the pattern classifiers are 100 μm . The rationale is detailed in the response to question 19.

24. The device architecture presented in the main text has two layers of metasurfaces. For more consistency, I recommend revising Fig. 1a accordingly.

Response: Thanks for the reviewer's suggestion. We have revised Fig. 1a according to the reviewers' suggestions.

25. According to the scale bar in Fig. 2e, it seems the length of some slots is $> 3.7 \mu\text{m}$? Is there any specific reason behind this, and how this affects the overall response? It would be good to consider this when generating Fig. 1c. Also, the width of slots seems to me $\sim 130 \text{ nm}$. This is not in accordance with the simulation results.

Response: We appreciate the reviewer's careful check. We have updated the SEM images and the scale bars.

Here is the revised Fig. 2:

26. Most figures can be regenerated with higher resolution and details.

Response: Thanks for the suggestions. We updated the figures with a higher resolution.

27. Among minor concerns, there are some (though not many) typos and poorly formulated sentences across the manuscript:

* Page 2: “Both amplitude modulation the phase shift”, “Fig. 1a-b”, “meta-system”, “represents a weight element and connecting”

* Page 3: “Fig. 1c-d”, “gradient descent algorithm”

* page 4: (Fig. 3a) is duplicated in one sentence, Fig. 2c should be revised as Fig. 2b, “With an input image of letter ‘X’, the light intensity onto channel 1 (Ch1) on the output plane”, “The light intensity the position of Ch1”, “10,000 such datasets”, “1000 testing dataset”, “and experimentally verified scanning the input”

* page 5: “by a designing one for”

* Page 6: “complexed”, “in such system”, “device layer”, “e.g.”

* Page 10: “coupled onto”

Response: We are thankful for the reviewer’s careful check. We have updated those terms in the paper and proofread the rest of the contents.

REVIEWERS' COMMENTS

Reviewer #1 (Remarks to the Author):

The authors have replied all my comments and I don't have any further comments. I think this paper could now be accepted by Nature Communications.

Reviewer #2 (Remarks to the Author):

The authors have successfully addressed my previous comments. I would like to support its publication in NC.

Reviewer #3 (Remarks to the Author):

I am satisfied with the comprehensive/clear responses from the authors. I believe the manuscript is suitable for publication.